# Bridging the age gap in breast cancer: evaluation of decision support interventions for older women with operable breast cancer: protocol for a cluster randomised controlled trial

Karen Collins,[1] Malcolm Reed,[2] Kate Lifford,[3] Maria Burton,[1] Adrian Edwards,[3] Alistair Ring,[4] Katherine Brain,[3] Helena Harder,[5] Thompson Robinson,[6] Kwok Leung Cheung,[7] Jenna Morgan,[8] Riccardo Audisio,[9] Susan Ward,[10] Paul Richards,[10] Charlene Martin,[8,11] Tim Chater,[12] Kirsty Pemberton,[12] Anthony Nettleship,[13] Christopher Murray,[13] Stephen Walters,[12] Oscar Bortolami,[12] Fiona Armitage,[11] Robert Leonard,[14] Jacqui Gath,[15] Deirdre Revell,[15] Tracy Green,[15] Lynda Wyld[8,11]

For numbered affiliations see end of article.

**Correspondence to**
Lynda Wyld;
l.wyld@sheffield.ac.uk

## ABSTRACT

**Introduction** While breast cancer outcomes are improving steadily in younger women due to advances in screening and improved therapies, there has been little change in outcomes among the older age group. It is inevitable that comorbidities/frailty rates are higher, which may increase the risks of some breast cancer treatments such as surgery and chemotherapy, many older women are healthy and may benefit from their use. Adjusting treatment regimens appropriately for age/comorbidity/frailty is variable and largely non-evidence based, specifically with regard to rates of surgery for operable oestrogen receptor-positive disease and rates of chemotherapy for high-risk disease.

**Methods and analysis** This multicentre, parallel group, pragmatic cluster randomised controlled trial (RCT) (2015-18) reported here is nested within a larger ongoing 'Age Gap Cohort Study' (2012-18RP-PG-1209-10071), aims to evaluate the effectiveness of a complex intervention of decision support interventions to assist in the treatment decision making for early breast cancer in older women. The interventions include two patient decision aids (primary endocrine therapy vs surgery/antioestrogen therapy and chemotherapy vs no chemotherapy) and a clinical treatment outcomes algorithm for clinicians.

**Ethics and dissemination** National and local ethics committee approval was obtained for all UK participating sites. Results from the trial will be submitted for publication in international peer-reviewed scientific journals.

**IRAS reference** 115550.

**Trial registration number** European Union Drug Regulating Authorities Clinical Trials (EudraCT) number 2015-004220-61;Pre-results. Sponsor's Protocol Code Number Sheffield Teaching Hospitals STH17086. ISRCTN 32447*.

---

* The wider Age Gap study commenced as a cohort study in 2012/2013, collecting prospective observational data on older women. At the time there was no requirement

## Strengths and limitations of this study

► The two evidence-based decision support interventions (DESIs) for women over 70 years diagnosed with breast cancer who are offered a choice of primary endocrine therapy or surgery (plus adjuvant endocrine therapy, hereafter termed surgery/AET, antioestrogen therapy) or chemotherapy versus no chemotherapy are, to the best of our knowledge, the first of its kind worldwide.

► The web-based clinical outcomes management algorithm is the first of its kind and allows patient age, comorbidities, frailty and cancer characteristics to be considered in predicting breast cancer survival and cancer outcomes.

► A limitation of the trial will potentially be selection bias from recruitment and poor uptake/utilisation of the DESIs at intervention sites.

► A second limitation may be an inability to demonstrate a benefit in terms of cancer survival rates without at least 5–10 years follow-up or an overall survival advantage due to the competing causes of death in this age group.

## INTRODUCTION
### Background and rationale
Breast cancer is the most common cancer in

for registration on the ISRCTN database as the trial was approved prior to 2013 and was only a cohort study, therefore the study team made public notification via the Cancer help database and more recently registered it on the EURDRACT database last year. The trial protocol was changed late 2015/2016 to convert the study to a cluster RCT and at that point registered the revised protocol with the ISRCTN.

women in the UK, with over 53 000 new cases being diagnosed in the UK each year.[1] Of these, 16 000 women will be over the age of 70, a figure which is rising steadily as the UK population ages.[2] While breast cancer outcomes are improving steadily in younger women due to advances in screening and improved therapies, there has been little change in outcomes in this older age group of women. The UK lags significantly behind other European countries in its outcomes for these women. There is a wide variation in practice in the management of breast cancer in older women.[3] The gold standard of care for early breast cancer is surgical removal of the primary cancer (mastectomy or conservation surgery), and diagnostic or therapeutic axillary nodal surgery followed by stage and immunophenotype appropriate adjuvant therapies (chemotherapy, trastuzumab, antioestrogens and radiotherapy) to reduce the risks of disease recurrence. There is consistent evidence that older women are less likely to receive surgery, chemotherapy, radiotherapy and trastuzumab, based on the premise that there is less evidence of efficacy and a greater risk of treatment morbidity.[4] In the case of surgery, up to 40% of older women do not undergo surgery for their breast cancer, and their treatment is mainly with antioestrogen tablets alone, known as primary endocrine therapy (PET).[5] While it is inevitable that in older women, comorbidities and frailty rates are higher, and which will increase the risks of some breast cancer treatments, such as surgery and chemotherapy, many older women are healthy and will benefit in terms of breast cancer outcomes from their use. Selection of appropriate age, comorbidity and frailty-adjusted treatment regimens is highly variable, largely non-evidence based and often fails to adequately consider the needs or wishes of patients. Two key areas of local practice variation are rates of surgery for operable oestrogen receptor (ER) positive disease and rates of chemotherapy for high-risk disease. PET rates vary fourfold between UK centres[3] and are not accounted for by case mix adjustment. Similarly, rates of chemotherapy vary 10-fold.[4]

Recent reports have advocated the use of PET only in the very old or frail.[6] Current national guidelines state that patients with operable breast cancer should be treated with surgery, and not PET, 'irrespective of age' unless this is precluded by comorbidities[7]; while the International Society of Geriatric Oncology and European Society of Breast Cancer Specialists recommend that PET should only be offered to patients with a 'short estimated life expectancy (less than 2 to 3 years), who are considered unfit for surgery… or who refuse surgery.'[8] However, as a large number of older women are treated with PET in UK and other countries, it is not clear whether this guidance is being followed consistently. PET is associated with high rates of patient satisfaction and low treatment morbidity but in the medium and long term some women may need a change of therapy once antioestrogen resistance develops.[9] Randomised trials and a recent Cochrane review have shown that surgery (plus adjuvant antioestrogens hereinafter termed surgery/

AET, antioestrogen therapy) and PET have equivalent overall survival rates.[10 11] However, for fitter women with a longer predicted life expectancy, there is evidence that breast cancer specific survival rates are inferior with PET.[12] For very frail women where surgery would be unsafe or poorly tolerated, PET is the clear choice in women with oestrogen-sensitive disease.[12]

For women at intermediate or higher risk of surgery complications, there is a complex series of trade-offs to be made for each patient. The decision must balance the risks of surgical morbidity (pain, risks associated with hospitalisation, surgical complications) but with a greater certainty of local disease control, against the minimal morbidity with PET but a risk of later local disease progression and the need for a change of treatment to either surgery or alternate AET.[13–15]

Chemotherapy utilisation is also very low in women over 70 (14%)[4] and almost non-existent in women over 80, even in those where high phenotypic risk is present (high grade, node positive, ER negative, HER2 positive).[4] Rates of chemotherapy can vary widely between UK breast units, between 6% and 60% in high-risk women.[16] This reflects the fact that most of the randomised trials have upper age cut-offs at age 70 or recruit very poorly in this age group, meaning there is little evidence of whether it is effective or not. In addition, there is evidence of an increased risk of significant complications such as neutropenic sepsis in older women.[17] This clearly suggests that guidelines for best practice are required. The primary tool used by oncologists to determine the likely benefit of chemotherapy on a patient level basis is Adjuvant!Online,[18] although this has been shown to be inaccurate in older women.[19] The more recently developed PREDICT tool[20] performs better in this age group but has limited functionality for taking comorbidity and frailty into account.

This cluster randomised trial will evaluate the implementation of two ('complex') decision support interventions (DESIs) designed to be used by both clinicians and patients to assist in the decision making about treatment for early breast cancer in older women.

### The Bridging the Age Gap study

The Bridging the Age Gap study[21] is an NIHR (National Institute for Health Research) funded programme of research (2012-18RP-PG-1209-10071) examining breast cancer management in older women with the ultimate aim of improving outcomes by providing high-quality evidence to support treatment decision making in this age group.

The study protocol reported here focuses exclusively on the cluster randomised trial part of the wider Bridging the Age Gap study.[21] The study group has developed two patient-facing DESIs based on a systematic evidence summary, expert reference group consultation, patient interviews[22–24] and questionnaires about informational needs and preferences and extensive user and field testing with both healthy older women and older

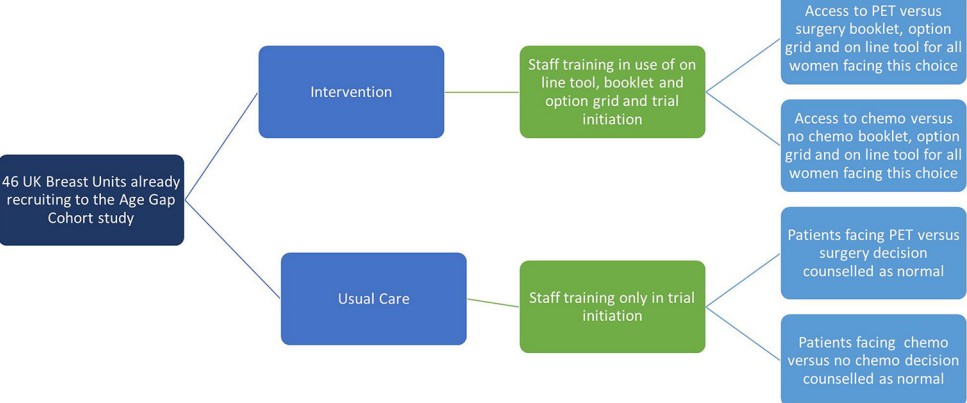

**Figure 1** Overview of the cluster randomised controlled trial. PET, primary endocrine therapy.

women who had faced the decision relating to the choice of surgery/AET or PET in frailer women with ER-positive breast cancer, and the decision regarding use of adjuvant chemotherapy in fitter women with high-risk cancers. Each DESI includes a clinician-facing clinical management algorithm and two patient-facing decision aids (PtDAs). The clinician-facing management algorithms derive from detailed cancer registry outcome data linked to treatment-related morbidity and patient and cancer characteristics from the UK cancer registry (2002–2010) for two UK regions (Northern and Yorkshire and East Midlands) which are representatives of the UK population as a whole in terms of demography, population structure and deprivation. This is a large diverse area, representing 23% of the UK population.[25] These online algorithms allow patient age, comorbidities, frailty and cancer characteristics to be considered by a clinician in predicting survival and cancer outcomes and to help inform breast cancer management decisions for older women.[25] The PtDAs are in the form of a booklet and a (brief) option grid for the clinical decision in question.[26 27]

The trial will evaluate these tools in a cluster randomised trial across 53 UK breast units according to the study schematic (figure 1).

The aim of this trial is to evaluate if, how and to what extent, the use of the DESIs embedded as 'standard of care' within intervention arm sites improves quality of life (QoL), decision quality (integrating knowledge, attitudes and decision made), coping and illness representations, and reduces decision regret, thus indicating improved informed decision making of older women about treatment options for their breast cancer.

To our knowledge, this is the first randomised controlled trial (RCT) to have been undertaken to explore this issue.

### Objectives
The objectives are:
1. To assess the effectiveness of the implementation of DESIs[26 27] in clinical practice in terms of improving patient QoL, decision quality (integrating knowledge, attitudes and decision made), coping and illness representations, and reducing decision regret, thus indicating improved informed decision making.
2. To determine if how, or to what extent, the clinical outcomes management algorithm impacts on clinical decision making among clinicians (change in PET/surgery rates and chemotherapy rates).
3. To determine whether the DESIs are effective in improving short, medium and long-term cancer outcomes in this age group of women (treatment morbidity and overall and disease-specific survival).
4. To assess the utility and uptake of the DESIs from the perspective of both clinicians and patients by undertaking a formal process evaluation.

### Hypotheses
1. Use of the DESIs will improve the QoL in older women with operable breast cancer and ultimately improve cancer outcomes.
2. Older women faced with a choice of treatment decisions for their breast cancer will report an improved decision quality and shared decision-making experience and less decision regret using DESIs compared with older women who receive usual clinical decision-making support.
3. Use of evidence-based DESIs will improve short and longer term outcomes by improving treatment personalisation to a woman's health, fitness and cancer characteristics and by improving the quality of decision making, and reduce the heterogeneity of practice across the UK.
4. Women in the intervention sites will express more positive illness representations (eg, increased personal control, positive emotional consequences, less overall threat) and increased use of engagement coping strategies compared with women from the control sites.

## METHODS

### Study design and setting

This protocol follows the CONSORT (Consolidated Standards of Reporting Trials) statement guidelines for cluster trials.[28]

This study is a multicentre, parallel group, pragmatic cluster RCT (2015-18).[29] It is nested within a larger ongoing Bridging the Age Gap cohort study (2012-18)[21] (figure 1) which is currently recruiting from 53 breast units within in the UK (observational cohort study of current UK management of older women with early breast cancer).

### The RCT study

The intervention comprises implementation of a package of two DESIs for the PET versus surgery/AET, or chemotherapy versus no chemotherapy decisions. Each DESI includes an online algorithm for treatment outcomes, and two PtDAs—a booklet and a brief option grid.[26 27] Each DESI is a complex intervention, including training for the clinician (breast surgeon, medical oncologist, breast care nurses) on the use of the algorithm (surgeons and medical oncologists only) or PtDAs, and the clinician and patient decide which, if any, of these elements they wish to use to assist the decision-making process, the intention being for the intervention to be used as part of everyday clinical practice/pathway within the intervention sites.

Each online algorithm includes functionality to adjust outcome prediction according to patient age, comorbidity, frailty, tumour stage and ER status and which gives outputs of 2 and 5-year overall and breast cancer specific survival. The algorithms were developed in the earlier phase of the Age Gap study[25] and were designed to guide clinicians and their patients in the treatment of:

1. frailer older women with ER-positive breast cancer to optimise treatment with either PET or surgery/AET;
2. fitter older women who have already had primary surgery and been found to have high-risk cancer characteristics (eg, ER-negative, HER2-positive or node-positive breast cancer) to optimise treatment with either adjuvant chemotherapy or no adjuvant chemotherapy (note the term chemotherapy includes chemotherapy +/− trastuzumab if appropriate).

The algorithm is based on a computer model of predicted outcomes and variance caused by patient and disease parameters. Unlike existing web-based algorithms for cancer treatment (Adjuvant!OnLine[19] or PREDICT[20]) which do not have the facility to specify frailty or comorbidity in detail (or at all), the Age Gap algorithm permits these factors to be taken into account. The Age Gap tool has been optimised for accuracy in this age group and has been based on analysis of data from over 20 000 UK women over the age of 70 derived from cancer registry data. The algorithm has built-in educational materials (including several online presentations, data sources, FAQs and an animated educational video). The online algorithm is designed to be used by clinicians to guide treatment decision making and its outputs can be printed off in a patient-facing format that could be used in personalised patient counselling. The report provides specific survival estimates for each treatment option for an individual woman based on her personal and cancer characteristics. This works in much the same way as the printouts from Adjuvant!Online[19] or PREDICT[20] but in this case developed for the PET versus surgery/AET decision and with more detailed data entry relating to the woman's age and fitness level.

Two PtDAs (PET vs surgery/AET[26] and chemotherapy vs no chemotherapy[27]) have been developed during the earlier phase of the study.[22–24] The PtDAs comprise an option grid[30] and a booklet for each decision. The option grid is a one-page evidence-based summary of the treatment options alongside patients' frequently asked questions, helping patients to differentiate the key features, risks and benefits of treatment options in relation to their personal values and preferences. The option grid has been designed to be sufficiently brief for use in clinical encounters and accessible enough to support a better dialogue between patients and their clinical team.[30] The booklet provides information about both options including diagrams, side effects and potential risks and benefits. It also includes a section to guide deliberation and encourage the patients to clarify their preferences based around identifying 'what is most important to them.'[16]

### Eligibility criteria

#### Inclusion criteria

1. female;
2. aged over 70 years at the time of diagnosis of cancer;
3. primary operable (tumour, node, metastasis (TNM) categories V7: T1, T2, T3, N0, N1, M0), ER-positive invasive breast cancer (core biopsy or diagnostic incision biopsy);
4. ability to give informed consent and to read English.

#### Exclusion criteria

1. disease unsuitable for surgery, for example, inoperable, locally recurrent or metastatic disease;
2. previous invasive breast cancer within the last 5 years;
3. non-English speakers.

### Data collection and outcomes

#### Primary outcome measure

The primary outcome measure for the RCT is global health status/QoL score (questions 29+30 only of The European Organization for Research and Treatment of Cancer QLQ-C30 Reference Manual) (EORTC QLQ-C30).[31] This primary endpoint was stipulated by the funder of the study with the justification being that the EORTC QLQ-C30 is internationally recognised and well-validated QoL measure (as opposed to our original primary endpoint of decision quality). This was measured at 6 weeks and 6 months postdiagnosis/consent.

An independent data monitoring committee (DMC) comprising three experienced academic clinicians

oversees the study and monitors trial conduct and safety and potential harm and has access to all study data; the role being to provide recommendations for trial changes (or closure). Data collection is being undertaken by trained clinical staff within each of the participating sites. The study data manager and study monitor also undertake regular site visits to outline the study protocol, ensure protocol adherence and monitor data collection and completeness. Data collection for the study includes detailed information about the patients and their cancer at the time of diagnosis: age, comorbidity (Charlson comorbidity index,[32] frailty—The Barthel Index (ADL, activities of daily living)[33] and instrumental activities of daily living scores (IADL)[34]), cognitive status (Mini-Mental State Examination),[35] baseline QoL (EORTC QLQ C30,[31] EORTC breast cancer-specific QoL questionnaire (QLQ-BR23),[36] EORTC QoL questionnaire module for older people with cancer (QLQ-ELD14),[37] EuroQol Group EQ-5D[38]), tumour stage, grade and receptor status. Treatment details are recorded including the type of surgery to the breast and axilla, use of adjuvant therapies (chemotherapy, radiotherapy, trastuzumab and hormonal therapies), including doses and adverse effects recorded using the Common Terminology Criteria for Adverse Events grading system. Follow-up is at baseline, 6 weeks, and 6, 12, 18 and 24 months after diagnosis/consent. Cancer outcomes, QoL and adverse events are recorded at each visit and in the longer term, women are asked to sign a consent form to permit the trial to collect their Cancer Registry data which will be collected 5 and 10 years following diagnosis and consent to the study. These data will permit us to look at whether using the DESIs alters patterns of treatment decision making between control and intervention sites and whether these impact on long-term outcomes. As such this is a uniquely detailed evaluation of such DESIs.

In addition, specific questionnaires relating to patient choice and decision making will be administered. These will apply to all women offered a choice of either PET and surgery/AET or chemotherapy versus no chemotherapy and are administered in relation to the time of their treatment choice. Secondary outcome measures here include decision regret (Decision Regret Scale,[39] shared decision making (CollaboRATE[40]), patient anxiety (Spielberger short-form state scale of the State-Trait Anxiety Inventory,[41] knowledge and preference (knowledge, readiness to decide and preference measure[42 43]), illness perceptions (Brief Illness Perceptions Questionnaire[44]) and Coping (brief COPE, Coping Orientation to Problems Experienced)[45]). Original data collected are entered and kept on file within each of the study sites. These data are entered electronically and stored securely onto password-protected databases within local databases and the main trial office. All records that contain names or other personal identifiers, such as locator forms and informed consent forms, are stored separately from study records identified by code number. Only the study steering and DMC have access

to the full trial data set. Errors, discrepancies or missing data are captured by the computer programme and the study data manager checks and subsequently follows this up with participating sites.

The timescales for each of these are shown in table 1.

## Sample size calculation

The primary endpoint will be the global health status/QoL scale (questions 29 and 30 of the EORTC-QLQ-C30)[31] at 6 months post baseline. Assuming an SD of 21 points for the global health status/QoL scale and a mean difference of 7 or more points on the global health status/QoL scale between the groups is of clinical/practical importance (a 'small' standardised effect size of 0.33). With no allowance for clustering; for the PET versus surgery DESI comparison with 190 eligible women per group we will have a 90% power of detecting this difference or more as statistically significant between the groups at the 5% two-sided level. If we assume an intraclass correlation of 0.03 then allowing for the clustered RCT design we will need to recruit 10 women, eligible for using the decision aids, per cluster (ie, 50 clusters × 10 women), 500 in total (this assumes a design effect of 1.3). With a 20% loss to follow-up by 6 months we need to recruit 13 women per cluster (50 clusters × 13 women) or 650 in total (325 per group).

## Randomisation

Randomisation is at breast unit level, stratified by high and low PET and chemotherapy rates. It was therefore not possible to blind the investigators or the study sites to the allocation of participants. Data for this stratification have been derived from the wider cohort study which has collected data on treatment rates for both PET versus surgery/AET and chemotherapy versus no chemotherapy.

## Control arm

Usual standard practice for older women (>70 years) diagnosed with breast cancer with no change to normal treatment decision-making practice.

## Intervention arm

Usual standard practice for older women (>70 years) diagnosed with breast cancer plus optional clinician and patient access to the package of DESIs which will have been made available to these units to adopt as their standard of care.

In the run into the trial period (June–December 2015), clinical teams (clinicians, research and breast nurses) from the participating sites attended a training event to enhance concordance with the study protocol (control group) and provide additional training on shared decision making and the use of the DESIs (intervention group). This comprised a 2-hour practical workshop which consisted of presentations, demonstrations and discussion based on the MAGIC programme.[46]

## Recruitment

Potentially eligible women are identified by clinicians (breast surgeons, medical oncologists and specialist breast nurses) and research nursing staff within

| Table 1 | Questionnaire schedule | | | | | |
| --- | --- | --- | --- | --- | --- | --- |
| **Standard age gap questionnaires** | **Baseline** | **6 weeks** | **6 months** | **12 months** | **18/24 months** | **Long term** |
| IADL | * | | | | | |
| ADL | * | | | | | |
| MMSE | * | | | | | |
| ECOG performance status | * | | | | | |
| Subjective global assessment | * | | | | | |
| Comorbidity | * | | | | | |
| EQ5D | * | * | * | * | * | |
| QoL (EORTC-QLQ C30, QLQ-BR23 and QLQ-ELD14) | * | * | * | * | * | |
| Decision quality | * | * | | | | |
| RECIST if PET | * | * | * | * | * | |
| Registry data access | | | | | | * |
| Tissue access | * | | | | | * |
| Tumour details | * | | | | | |
| Treatment details | * | * | * | * | * | |
| Adverse events | * | * | * | * | * | |
| **New for DESI study** (if offered choice of either pet or surgery/AET, or chemotherapy/no chemotherapy) | **Baseline** (after consent for PET or surgery) (AET or after consultation for chemo/no chemo, as applicable) | **6 weeks after relevant treatment choice** | **6 months after relevant treatment choice** | | | |
| Spielberger anxiety | | * | * | | | |
| Collaborate | * | | | | | |
| Decision regret | | * | * | | | |
| Knowledge readiness to decide and preference measures | * | | | | | |
| Brief IPQ | | * | * | | | |
| Brief COPE | | * | * | | | |
| Process evaluation (if taking part in process evaluation) | | | | | | |
| Process evaluation questionnaire | | * | | | | |

*indicated that these data are collected at this time point.
ADL, activities of daily living; AET, antioestrogen therapy; COPE, Coping Orientation to Problems Experienced; DESI, decision support intervention; ECOG, Eastern Cooperative Oncology Group; IADL, instrumental activities of daily living; IPQ, Illness Perception Questionnaire; MMSE, Mini-Mental State Examination; PET, primary endocrine therapy; QoL, quality of life; RECIST, Response Evaluation Criteria In Solid Tumors.

multidisciplinary teams of the study sites. Study packs are being given to eligible patients either following their clinical consultation where either PET or surgery/AET options or chemotherapy versus no chemotherapy options are discussed. Monthly study newsletters are sent to all participating sites to provide feedback to staff in order to maintain interest and recruitment to the study. Any modifications to the original study protocol will be discussed with the data monitoring and ethics committee and approvals sought from the funder and the ethics committee. Recruitment for the trial has now commenced and 750 women have been recruited over the 53 participating sites.

### Data analysis
The statistical analyses will be performed on an intention-to-treat basis comparing the DESI and control groups. All statistical exploratory tests will be two tailed with p=0.05. Baseline demographics (eg, age), physical measurements and health-related QoL data will be assessed for comparability between the treatment groups. A marginal generalised linear model (GLM), with coefficients estimated using generalised estimating equations (GEE) with robust SEs and an exchangeable autocorrelation matrix in STATA will be used to analyse the outcomes and allow for the clustered nature of the data. The exchangeable correlation structure corresponds to an equal correlation

model, meaning that the correlations of the outcomes with a cluster, that is, breast centres, are the same. For continuous outcomes, such as mean global health status/QoL score (questions 29+30 of EORTC QLQ-C30[31]) at 6 months postdiagnosis/consent intervention, knowledge score and preference for treatment score, an identity link with a normal distribution for the outcome will be used. Estimates for the treatment group coefficient from this regression model will be reported along with their associated 95% CI. In the event of differences between the intervention and control groups with respect to baseline demographic, physical and health-related QoL measurements, then these covariates will be used in the GLM to adjust the treatment effect for these variables. The adjusted regression coefficient estimate for the treatment group parameter along with its 95% CI will then be reported.

For the other secondary outcomes, at 6 weeks and 6 months, such as the other dimensions of the EORTC QLQ C30,[31] the EORTC QLQ-BR23[36] and EORTC QLQ-ELD14,[37] the mean QoL dimension scores will be compared between the intervention and control groups using similar models.

A series of exploratory subgroup analyses using a marginal GLM with coefficients estimated using GEE with robust SEs and an exchangeable autocorrelation matrix, with the primary outcome the mean global health status/QoL score (questions 29+30 of EORTC QLQ-C30[31]) at 6-month postdiagnosis/consent randomisation as the response will be carried out. An interaction statistical test between the randomised intervention group and subgroup to directly examine the strength of evidence for the treatment difference between the treatment groups (intervention vs control) varying between subgroups will be undertaken. Age subgroup (75–79, 80–84, 85–89 and 90+ years) and comorbidity levels (based on the modified Charlson comorbidity score[32]) will be the only a priori defined subgroups to be considered for interaction test. Subgroup analysis will be performed regardless of the statistical significance on the overall intervention effect (intervention vs control).

### Missing primary outcome data
A sensitivity analysis using a variety of imputation methods to impute any missing primary outcome data (6-month EORTC QLQ-C30[31] global health status/QoL score) will be performed. The imputation methods will include last observation carried forward, regression and multiple imputation. The estimates of the treatment effect and its associated CI, from the various imputation methods, will be graphically displayed alongside the results for the observed data.

### Process evaluation
Running alongside the main study, a detailed mixed methods process evaluation is being undertaken at 16 sites to assess the implementation of the DESIs (fidelity to the trial protocol) to consider the DESIs' usefulness and acceptability and examine the facilitators and barriers to embedding them into everyday clinical practice. A

random selection of breast units was made stratified by trial arm and recruitment rate to the cohort study (high/low PET/surgery/chemo rates).

In summary, the Age Gap study[21] aims at improving outcomes of older women diagnosed with breast cancer by providing high-quality evidence to support treatment decision making in this age group. The two evidence-based DESIs each include a clinical management algorithm and two PtDAs in the form of a booklet and a (brief) option grid for the clinical decision in question. These online algorithms will allow patient age, comorbidities, frailty and cancer characteristics to be considered by a clinician in predicting survival and cancer outcomes and to help inform breast cancer management decisions for older women.

**Author affiliations**
[1]Centre for Health and Social Care Research, Sheffield Hallam University, Sheffield, UK
[2]Brighton and Sussex Medical School, University of Sussex, Brighton, UK
[3]Division of Population Medicine, School of Medicine, Cardiff University, Cardiff, UK
[4]Royal Marsden Hospital, London, UK
[5]SHORE-C, University of Sussex, Falmer, Brighton, UK
[6]Department of Cardiovascular Sciences, Leicester Royal Infirmary, Infirmary Square, Leicester, UK
[7]School of Medicine, University of Nottingham, Royal Derby Hospital Centre, Nottingham, UK
[8]Department of Oncology and Metabolism, University of Sheffield, Sheffield, UK
[9]Department of Surgery, University of Liverpool, Liverpool, UK
[10]Department of Health Economics and Decision Science, School for Health and Related Research, University of Sheffield, Sheffield, UK
[11]University of Sheffield, Sheffield, UK
[12]Clinical Trials Research Unit, School for Health and Related Research, University of Sheffield, Sheffield, UK
[13]Department of Epigenesys, School for Health and Related Research, University of Sheffield, Sheffield, UK
[14]Imperial College Healthcare NHS Trust, London, UK
[15]Yorkshire and Humberside (formerly North Trent Cancer Network) Consumer Research Panel, Sheffield, UK

**Acknowledgements** The authors are grateful to all participating sites, and specifically site principal investigators who have or who are currently contributing to the RCT.

**Contributors** LW, KC and MR were overall project leads. Decision aid development and testing: KL, MB, KC, LW, MR, AE, KB, JG, DR, FA, HH. Trial management: LW, CM, TC, KP. Online tool development: LW, SW, PR, AN, CM, MB, JM. Statistical advice: SW, OB. Trial management group: All authors. Chemotherapy advisors: AR, RL, HH. All authors have contributed to reading and approved the final manuscript.

**Funding** This work was supported by the National Institute for Health Research Programme Grant (grant number RP-PG-1209-10071).

**Competing interests** None declared.

**Ethics approval** London South East NHS Research Ethics Committee 12/LO/1808.

**Provenance and peer review** Not commissioned; externally peer reviewed.

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
