## [Reviewer comments · BMJ Open]

ARTICLE DETAILS

TITLE (PROVISIONAL)	Bridging the Age Gap in Breast Cancer: Evaluation of decision support interventions for older women with operable breast cancer: protocol for a cluster randomised controlled trial.
AUTHORS	Collins, Karen; Reed, Malcolm; Lifford, Kate; Burton, Maria; Edwards, Adrian; Ring, Alistair; Brain, Kate; Harder, Helena; Robinson, Thomas; Cheung, Kwok; Morgan, Jenna; Audisio, Riccardo; Ward, Susan; Paul, Richards; Martin, Charleen; Chater, Tim; Pemberton, Kirsty; Nettleship, Anthony; Murray, Christopher; Walters, Stephen; Bortolami, Oscar; Armitage, Fiona; Robert, Leonard; Gath, Jacqui; Revell, Deirdre; Green, Tracy; Wyld, Lynda

VERSION 1 - REVIEW

REVIEWER	Gerrit-Jan Liefers Department of Surgery Leiden University Medical Center PO Box 9600, 2300 RC Leiden, The Netherlands None declared (for the statistical part of the study I feel not quite qualified)
REVIEW RETURNED	21-Dec-2016

GENERAL COMMENTS	This protocol describes a very interesting and novel study. Evidence in older women indeed is lacking while the numbers are steadily increasing. The study describes (as well as the ongoing Age-Gap study) is a courageous effort to improve the decision making process for older breast cancer patients. Although I fully support the initiative I have some concerns regarding the logistics and outcome parameters. I believe that the researchers have gone through considerable efforts to allow for maximum contrast between arms (i.e. cluster randomization) but I am not sure that the staff training during the run-in phase will not affect the way of counselling in the control arm. If so, that would dilute the measured gain in QoL. Also one can question quality of life endpoint at 6 months (versus for example one or two years). The choice for only questions 29 and 30 in the EORTC QLQ-30, in my opinion, is excellent. One could then also choose for example a Cantrills' ladder. A minor point: I believe that there is mistake in figure 2B (page 13)/. The way I read the yellow protion of the graph, the vast majority 'died of other causes'. Perhaps yellow and green are mixed up?
---

REVIEWER	Prof Andrew Baildam Queen Mary University London United Kingdom
REVIEW RETURNED	26-Dec-2016

GENERAL COMMENTS	This is a highly worked study that aims to improve the outcomes of older women diagnosed with breast cancer, by using intervention with information strategies. Outcomes presently are poor compared with the advances in outcomes made over the last two decades for younger women with breast cancer. It specifically addresses patient information and awareness: less clear is how the present bias of some clinicians away from surgery and chemotherapy for the older patient can be challenged.
--

REVIEWER	Fiona MacNeill and Marios Tasoulis Royal Marsden Hospital London UK
REVIEW RETURNED	30-Dec-2016

GENERAL COMMENTS	Bridging the Age Gap in Breast Cancer This is an important project as the management of breast cancer is becoming a public health issue. Optimizing the management of older women with BC is an important issue for first world countries with ageing populations: Breast cancer is very common in the older woman but for those women with multiple co-morbidities, treatment planning and decision making requires highly individualized, complex risk benefit assessments (often with little reliable data regarding natural life expectancy and cancer related survival) and ultimately truly shared decision making. Older women are underrepresented in trials evaluating breast cancer treatment (JCO Dec 19 2016, Freedman et al). We need to better understand why breast cancer outcomes in older women have not mirrored the improvements seen in younger women and explore tools that may optimize individual treatment planning. The authors have submitted the protocol for a cluster randomized controlled trial evaluating decision support interventions (DESI's) in 1500 older women with operable breast cancer. The primary outcome is Q of L with a number of secondary outcomes. This study protocol is linked to a larger cluster randomized 'Age Gap cohort study' which assesses the impact of two DESIs on breast cancer outcomes: one for primary endocrine treatment versus surgery followed by adjuvant endocrine treatment, and the second for chemotherapy versus no chemotherapy in high risk patients. Major comments  1. Sections of this protocol are confusing as it is difficult to understand if the authors are referring to the main study 'Age Gap' or the described sub-study 'Bridging the Age Gap in Breast Cancer': for example in the 'strength and limitations' section of the protocol it is not clear which study these refer to. The manuscript needs to be much clearer with regards to which objectives, outcomes and methodology are part of the the sub-study and which are part of the main study and the overlaps. 2. Could parts of the main study be appendixed rather than embedded in the current protocol for ease of understanding?
--

	3. It may be the following comments simply reflect my struggle to untangle the 2 studies, if so apologies. 4. The investigators have used the data from the UK Cancer Registry for two UK regions (Northern and Yorkshire and East Midlands) in order to develop the clinical management algorithms that would help in the prediction of survival and cancer outcomes and ultimately in the decision making process. However, how can the investigators prove that these data are representative of the whole UK population and thus the outcomes of the algorithm reproducible and generalizable? 5. The primary endpoint is global health status/QoL score assessed at 6 weeks and 6 months post diagnosis/consent. Is the 6 month period sufficient to capture QoL in patients undergoing surgery and/or chemotherapy? 6. Randomization of the main study is difficult to understand: Page 13, lines 14-20. The patient and the clinician decide which, if any, of the interventions will be used to assist the decision making process. Does this mean that they might choose NOT to use any of the recommended interventions? In that case the randomization is violated. Please clarify. 7. Page 20, lines 19-24. If the use of the DESIs is optional how will this group differ from the control group? Please clarify 8. Will all patients in the 'intervention arm' units be enrolled into the study? If not, this might result in selection bias. Minor comments 1. Page 8 Line 30 'ample evidence of benefit for chemotherapy in women under 70' needs to be referenced as it is in conflict with the next sentence. 2. There are several typos throughout the manuscript. 3. Manuscript needs to be reviewed and tweaked for clarity EG. Page 8 line 14 'For women at intermediate or higher risk of surgery' should presumably read 'For women at intermediate or higher risk of surgery' complications
--	--

REVIEWER	Katherine Crew Columbia University USA
REVIEW RETURNED	08-Mar-2017

GENERAL COMMENTS	The authors describe the study protocol for a cluster randomized controlled trial of decision support tools for breast cancer treatment decisions among elderly women. The authors should address the following comments: 1. What is the rationale for choosing quality of life at 6 weeks and 6 months post-diagnosis as the primary endpoint? 2. The patient decision aids appear to be hardcopy brochures. How will updates in breast cancer treatments be incorporated into these decision aids? How will tailoring of individual patient characteristics and tumor characteristics be incorporated into the decision support
--

	tools? 3. What type of healthcare providers are being targeted by the decision support tools? Medical oncologists, breast surgeons, etc.? The authors describe training sessions for the providers, but more details should be provided about how the decision aids will be integrated into clinic workflow. 4. If the trial has already opened to enrollment (2015-2018), the authors should provide an update on current recruitment and retention.
--	--

VERSION 1 – AUTHOR RESPONSE

Reviewer 1: I believe that the researchers have gone through considerable efforts to allow for maximum contrast between arms (i.e. cluster randomization) but I am not sure that the staff training during the run-in phase will not affect the way of counselling in the control arm. If so, that would dilute the measured gain in QoL. Also one can question quality of life endpoint at 6 months (versus for example one or two years).

Author response: In our original protocol we proposed undertaking training on use of the interventions in the intervention study sites only. However, the NIHR insisted that we undertook similar training in the control sites to reduce the risk of bias. They were concerned that by offering training only in intervention sites we would bias the outcomes by making staff more aware of the issue, regardless of the impact of the intervention. They therefore wanted us to introduce the same training, minus the actual intervention, at ALL sites so the effect of the intervention would be more apparent. We were careful that no specific information on the intervention was given as part of this general training with training days in 2 parts: standard for all sites and then only intervention sites given instruction in the use of the tools. We therefore feel that although the reviewer is correct that this training in the control sites may have had some impact on patient counselling, it reduces the risk of bias in the outcome of the study.

Reviewer 1: I believe that there is mistake in figure 2B (page 13). The way I read the yellow portion of the graph, the vast majority ‘died of other causes’. Perhaps yellow and green are mixed up?

Author response: The figure is correct. The reviewer rightly points out that the figure demonstrates that the vast majority of women will die of other causes and NOT their breast cancer irrespective of whether they received surgery or PET only. This is to be expected in a cohort of elderly women where we know that other cause mortality is the more significant cause of death by a considerable margin. In addition, as most of the women in this age group have ER +ve breast cancer which tends to have a long natural history, with the presently short follow up period of the study we would expect few breast cancer deaths in the study cohort, especially as the protocol specifies that they have operable (i.e. early stage) disease when they are recruited.

Reviewer 2: This is a highly worked study that aims to improve the outcomes of older women diagnosed with breast cancer, by using intervention with information strategies...less clear is how the present bias of some clinicians away from surgery and chemotherapy for the older patient can be challenged.

.Author response: The study aims to effect clinical change in a number of ways. The cohort study data, in which the cluster trial is nested, will provide a unique dataset of elderly outcomes with fitness and frailty and treatment correction to allow us to develop a complex management algorithm which will ultimately be available on line (like NHS PREDICT) and will help clinicians to make better quality evidence based risk adjusted decisions for women in this age group. The impact and uptake of a retrospective data based on line tool is being evaluated in the cluster trial but this will be updated with actual prospective data in 2 years time. The tool will be freely available and the wide adoption of the

trial across the UK, with 53 breast units taking part, plus the likely publications and presentation at UK breast meetings will hopefully increase the use of the tool. We also hope that the data from the cohort study, examining real world practice in elderly women will show what the outcomes of under and overtreatment are. The randomised trials that have looked at treatment omission (surgery versus PET, chemotherapy and radiotherapy) have shown no survival disadvantage from under treatment which is in stark contrast to the data from cohort studies, which all show a disadvantage. This is likely to be due to the tightly controlled selection criteria for RCTs, usually specifying high levels of fitness, good disease biology etc and hence are not truly representative.

Reviewer 3: Sections of this protocol are confusing as it is difficult to understand if the authors are referring to the main study 'Age Gap' or the described sub-study 'Bridging the Age Gap in Breast Cancer': for example in the 'strength and limitations' section of the protocol it is not clear which study these refer to. The manuscript needs to be much clearer with regards to which objectives, outcomes and methodology are part of the sub-study and which are part of the main study and the overlaps.
Author response: Apologies. We have revised the manuscript to differentiate between the main programme grant and the RCT which forms part of the wider programme grant. Clarification within the objectives and outcomes have been made and statements such as 'The protocol reported refers specifically to the RCT...' have been included to minimise any potential confusion

Reviewer 3: Could parts of the main study be appended rather than embedded in the current protocol for ease of understanding?

Author response: In order to provide the context to the RCT study, a brief description of the wider programme grant is needed as the developed DESIs and clinical algorithm were developed in earlier stages of the study. We do recognise this is a complex study and therefore we have revised the manuscript to differentiate between the main programme grant and the RCT which forms part of the wider programme grant.

Reviewer 3: The investigators have used the data from the UK Cancer Registry for two UK regions (Northern and Yorkshire and East Midlands) in order to develop the clinical management algorithms that would help in the prediction of survival and cancer outcomes and ultimately in the decision making process. However, how can the investigators prove that these data are representative of the whole UK population and thus the outcomes of the algorithm reproducible and generalizable?

Author response: The UK Cancer registry data for these 2 regions are representative of the UK population as a whole in terms of demography, population structure and deprivation. This is a large diverse area, representing 23% of the UK population and has some of the highest rates of data completeness and quality for breast cancer data as WMCIU is the lead registry for breast cancer. Data completeness is very high, at over 90%.

Reviewer 3: The primary endpoint is global health status/QoL score assessed at 6 weeks and 6 months post diagnosis/consent. Is the 6 month period sufficient to capture QoL in patients undergoing surgery and/or chemotherapy?

Author response: We would ideally like to capture longer term outcomes and will apply for funding to extend the study but at present this funding is not in place and so we cannot claim that this longer term outcome will be available. We will have longer term data on some of the patients but not all. If this is forthcoming, we will publish a revised outcome with longer term data

Reviewer 3: Randomization of the main study is difficult to understand: Page 13, lines 14-20. The patient and the clinician decide which, if any, of the interventions will be used to assist the decision making process. Does this mean that they might choose NOT to use any of the recommended interventions? In that case the randomization is violated. If the use of the DESIs is optional how will this group differ from the control group? Please clarify

Author response: Yes use of the tools is optional and indeed is one of the outcomes that will be

assessed as part of the process evaluation. This is a key aspect of the study. A tool may be highly effective but if it is not used in real world clinical practice it will have little impact. This will be explored within the protocol by a comparison of per protocol and intention to treat analyses as we capture whether the tools are used as part of the study data collection protocol

Reviewer 3: Will all patients in the 'intervention arm' units be enrolled into the study? If not, this might result in selection bias.

Author response: It is never possible to recruit all patients into any trial so there is always a risk of bias wherever consent needs to be taken. We will examine this with our comparison of study recruited cases with the UK population in terms of age mix, fitness and frailty mix and cognition levels, all of which are collected as part of the study. Data from the cohort study so far, where we have nearly 3000 recruits shows that our age mix, fitness and frailty mix etc are broadly similar to UK norms although a full detailed analysis has not yet been done.

Reviewer 3: Page 8 Line 30 'ample evidence of benefit for chemotherapy in women under 70' needs to be referenced as it is in conflict with the next sentence.

Author response: Apologies. This has been clarified to indicate there are very few clinical trials that have included participants over the age of 70.

Reviewer 3: There are several typos throughout the manuscript.

Author response: When the manuscript was converted into the PDF version for the online submission, it resulted in no spaces between words occurring in parts of the final manuscript. This has now been corrected.

Reviewer 3: Manuscript needs to be reviewed and tweaked for clarity EG. Page 8 line 14 'For women at intermediate or higher risk of surgery' should presumably read 'For women at intermediate or higher risk of surgery' complications

Author response: Done. This has been clarified to indicate there are very few clinical trials that have included participants over the age of 70.

Reviewer 4: What is the rationale for choosing quality of life at 6 weeks and 6 months post-diagnosis as the primary endpoint?

Author response: The NIHR mandated that we changed our original primary endpoint from 'decision quality/decision regret' to QoL at 6 weeks and 6 months post-diagnosis as a condition of funding.

Reviewer 4: The patient decision aids appear to be hardcopy brochures. How will updates in breast cancer treatments be incorporated into these decision aids? How will tailoring of individual patient characteristics and tumor characteristics be incorporated into the decision support tools?

Author response: The decision aids are provided in hardcopy to the patient by the clinician when treatment decisions are being made. The clinician inputs the patient's age, co-morbidities, frailty and cancer characteristics onto the online clinician algorithm which generates predicted survival and cancer outcomes in order to help inform breast cancer management decisions for a patient. This algorithm can then be printed off by the clinicians and given to the patient to add to their hardcopy decision aids.

Reviewer 4: What type of healthcare providers are being targeted by the decision support tools?

Medical oncologists, breast surgeons, etc.? The authors describe training sessions for the providers, but more details should be provided about how the decision aids will be integrated into clinic workflow.

Author response: Apologies. We have clarified this within the main text. i.e. breast surgeons, medical oncologists and breast nurses were given training on the interventions. However the clinical algorithm was to be used only by the breast surgeon or medical oncologist.

Reviewer 4: If the trial has already opened to enrolment (2015-2018), the authors should provide an update on current recruitment.

Author response: Thus far 750 patients have been recruited We have now incorporated this into the revised manuscript.

VERSION 2 – REVIEW

REVIEWER	Gerrit Jan Liefers Department of Surgery Leiden University Medical Center Leiden, The Netherlands
REVIEW RETURNED	14-Apr-2017

GENERAL COMMENTS	Important study in a growing population.
--

REVIEWER	Katherine Crew Columbia University, USA
REVIEW RETURNED	14-Apr-2017

GENERAL COMMENTS	The authors state QOL was chosen as the primary endpoint as recommended by the funding agency. However, they should still include the rationale for choosing this endpoint in the text of the manuscript.
---

VERSION 2 – AUTHOR RESPONSE

Reviewer: 4

The authors state QOL was chosen as the primary endpoint as recommended by the funding agency. However, they should still include the rationale for choosing this endpoint in the text of the manuscript.

Authors response: The quality of life endpoint was the primary endpoint stipulated by the NIHR. The justification being that they wanted a well validated outcome quality of life measure rather than a less well recognised softer end point like decision regret or collaborate. We have added this justification within the revised manuscript.

We hope that we have now incorporated an all revisions requested and it is not suitable for publication.

Best wishes Karen (Collins) on behalf of the named co-authors